# Relationship between Urinary Metabolomic Profiles and Depressive Episode in Antarctica

**DOI:** 10.3390/ijms24020943

**Published:** 2023-01-04

**Authors:** Kazuhiko Kasuya, Satoshi Imura, Takashi Ishikawa, Masahiro Sugimoto, Takeshi Inoue

**Affiliations:** 1Department of Gastrointestinal Surgery, Tokyo Medical University, Tokyo 160-0023, Japan; 2National Institute of Polar Research, Tokyo 190-8518, Japan; 3Department of Breast Oncology and Surgery, Tokyo Medical University, Tokyo 160-0023, Japan; 4Institute of Medical Sciences, Tokyo Medical University, Tokyo 160-0022, Japan; 5Institute for Advanced Biosciences, Keio University, Yamagata 997-0052, Japan; 6Department of Psychiatry, Tokyo Medical University, Tokyo 160-0023, Japan

**Keywords:** Antarctic regions, biomarkers, mental disorders, metabolomics, urine

## Abstract

Antarctic expeditions have a high risk of participant depression owing to long stays and isolated environments. By quantifying the stress state and changes in biomolecules over time before the onset of depressive symptoms, predictive markers of depression can be explored. Here, we evaluated the psychological changes in 30 participants in the Japanese Antarctic Research Expedition using the Patient Health Questionnaire-9 (PHQ-9). Urinary samples were collected every three months for a year, and comprehensive urinary metabolomic profiles were quantified using liquid chromatography time-of-flight mass spectrometry. Five participants showed major depressive episodes (PHQ-9 ≥ 10) at 12 months. The urinary metabolites between these participants and the 25 unaffected participants were compared at individual metabolite and pathway levels. The individual comparisons showed the most significant differences at 12 months in 14 metabolites, including ornithine and beta-alanine. Data from shorter stays showed less significant differences. In contrast, pathway and enrichment analyses showed the most significant difference at three months and a less significant difference at longer stays. These time transitions of urinary metabolites could help in the development of urinary biomarkers to detect subjects with depressive episodes at an early stage.

## 1. Introduction

People are resistant and adaptable to various environmental conditions. In particular, expeditioners in the Antarctic are exposed to considerable stress through isolation and extreme physical conditions [1], leading to unique psychological effects. In addition to the cold, danger, and hardship, the difficulties associated with individual adjustment to the group constitutes a major stressor [2], with members of Antarctic expeditions undergoing psychological changes resulting from exposure to long periods of isolation and confinement [3]. The psychological effects of polar expeditions in patients under abnormal light, cardiopulmonary, and cold conditions have been frequently investigated [3] since the first studies were performed in Polar Regions [2]. Such patients experienced alterations in peripheral circulation along with hypothermia and frostbite, in additional to immune system and hormonal abnormalities. Their symptoms included somatic and gastrointestinal complaints, fatigue, headaches, disturbed sleep, impaired cognition, depressed mood, anger and irritability, anxiety, and interpersonal tension and conflict [3]. Approximately 5% of the South Pole expedition participants fulfilled the criteria for psychiatric disorders as defined in the Diagnostic and Statistical Manual of Mental Disorders IV (DSM-IV) [4,5]. Therefore, the appropriate assessment of psychological status is necessary to prevent these issues.

The identification of psychological risk factors that have a negative effect on mood and behavior is desirable for designing efficient preventive approaches to reduce symptoms and generate satisfactory adaptation [6]. Seasonal patterns in depressive symptoms, such as winter-over syndrome, polar T3 syndrome, and subsyndromal seasonal affective disorder, have been identified [7,8,9]. The psychological profiles of the Japanese Antarctic Research Expedition (JARE) team over three consecutive years showed psychological changes at the end of winter, such as an increase in the planning trial score and a decrease in the body coldness score, despite showing high robustness against stress [10]. Psychological changes in the JARE team evaluated using a psychological scale based on an international comparison project of psychology called the Polar Psychology Project indicated that morale was high at the beginning of the winter season, fell into a midwinter slump during the polar night, and then recovered [11]. Together, these research reports reveal that not only assessing psychological changes but also appropriate early care is necessary to reduce the effects of psychological risks during Antarctic expeditions.

Molecular biomarkers have been identified to predict and quantitatively assess altered mental and immune status. A long-term stay in a confined environment increases stress; therefore, the development of objective molecular markers to assess mental health is required. An analysis of the changes in already established stress biomarkers, such as cortisol and dehydroepiandrosterone (DHEA), in researchers at Antarctic stations revealed that DHEA levels could predict mental health status [12]. The profiling of thyroid-stimulating hormone (TSH), melatonin, and cortisol during the Antarctic expedition suggested that different luminosity and temperature conditions increase psychological stress [13]. The profiling of blood cytokine and interleukin levels showed reduced T-cell function [14]. Altered cytokine production in the blood has also been observed [15,16]. Moreover, total serum thyroxine levels are significantly associated with tension anxiety, depression, and total mood disturbances [17]. Nevertheless, the evaluation of novel biomarkers to determine the association with their mental health in the extreme conditions associated with Antarctic expeditions remains necessary.

Recent omics technologies, which enable the simultaneous identification and quantification of thousands of molecules, have been used to explore new biomarkers associated with mental health. Among molecular profiling approaches, the large-scale analysis of metabolites via metabolomics has been recently used for biomarker discovery in the psychiatry field [18]. Owing to the large diversity of the chemical features of metabolites, a single approach cannot profile all metabolites; thus, the combination of individual separation systems prior to detection has been applied. For example, gas chromatography–mass spectrometry (GC/MS) was used to profile urinary metabolites to predict postpartum [19] and poststroke depression [20]. A combination of GC/MS and nuclear magnetic resonance (NMR) has been used to explore urinary metabolite biomarkers for major depressive disorder (MDD) and bipolar disorder (BD) [21,22,23]. Capillary electrophoresis-MS and liquid-chromatography-MS (LC-MS) were used to profile plasma hydrophilic metabolites in patients with MDD [24,25]. However, these studies were designed as cross-sectional studies, whereas a longitudinal study is necessary to evaluate the trend of changes in biomarker levels accompanied by mental status in order to realize the early identification of mental changes.

In this study, we aimed to elucidate the urinary metabolite biomarkers that have potential to discriminate subjects with depressive episodes from those without these symptoms. Here, we investigated the time course of urinary metabolites in the Japanese Antarctic Expedition team and evaluated their association with depressive episodes. Samples were collected every three months over a one-year period in Antarctica. Comprehensive metabolomic analyses were conducted with LC-MS. The individual metabolites and pathway-level changes associated with the transition of depressive episodes were identified.

## 2. Results

Thirty participants were enrolled in the study (Table 1). Figure 1 shows the time-series changes in the Patient Health Questionnaire-9 (PHQ-9) scores. Five participants showed a high PHQ-9 score (≥10) at the end of the 12-month stay, and were classified as the depressive episode group. The remaining 25 participants were included in the nondepressive episode group. At baseline (0-month stay), the PHQ-9 values were not significantly different between the two groups. However, after the third month of stay, a significant difference was observed between the groups (*p* < 0.05, Mann–Whitney U-test). Subsequently, the PHQ-9 values of the depressive episode group increased monotonically and were significantly higher at 9 and 12 months (*p* < 0.0001).

Hydrophilic urinary metabolites were identified and quantified using LC-MS-based metabolomic analysis. In total, 79 metabolites were frequently detected (≥95% of all samples), and 77 metabolites were used for subsequent analyses after excluding two insufficiently separated metabolites in LC analysis. The deviation of the peak areas among the quality control samples was 9.97 ± 7.09%. Score plots of the principal component analysis were used to reveal the overall similarity and discrepancy of the quantified metabolite patterns among the samples (Figure A1). The score plots for the samples collected at baseline were higher than those of the other samples. This indicates a large change in the metabolite concentration patterns of the baseline samples. However, these changes are considered to be influenced by differences in the sampling conditions and storage, as only the baseline samples were collected in Japan. Therefore, baseline data were excluded from subsequent analyses, and time-series data for the remaining one-year stays in Antarctica were compared.

Metabolites with significant differences (*p* < 0.05, Mann–Whitney test) of at least one point among the four data points (3- to 12-month stays) between the depressive and nondepressive episode groups were depicted in a heatmap (Figure 2). The time-course data are shown in Figure 3. Most metabolites exhibited significant differences in the data of the 12-month stay, including 2-aminoadipate, 3-methylhistidine, 4-guanidinobutanoate, alpha-aminobutyrate, beta-alanine, carnosine, cystathionine, cytosine, N-acetylhistamine, ornithine, and pyridoxamine. All of these metabolite concentrations were significantly lower in the depressive episode group compared with those of the nondepressive episode group at the 12-month stay (Figure 3a). Among these, only 4-guanidinobutanoate showed a significant difference in the false discovery rate (FDR)-adjusted *p* < 0.05. In addition, 2-aminoadipate levels were significantly lower in both the 9- and 12-month stays (Figure 3b). Guanine, putrescine, and indoleacetate showed significant differences at different time points during the 12-month stay.

Pathway and enrichment analyses were also performed between the depressive episode and nondepressive episode groups to evaluate the altered pathways. The results of the pathway analysis are shown in Figure 4. Each plot represents a single pathway. A higher Y-value (−log10(P)) indicates a smaller *p*-value, which means that more alternations were present at the pathway level. The higher X-axis value (pathway impact) indicates that the alteration of the mapped metabolites affects the pathway more widely. Therefore, pathways with higher values on both the X- and Y-axes were more completely altered. The three-month-stay data included five pathways that were significantly altered (*p* < 0.05): (1) nitrogen metabolites; (2) pyrimidine metabolism; (3) glyoxylate and dicarboxylate; (4) D-glutamine and D-glutamate metabolism; and (5) alanine, aspartate, and glutamate (Figure 4a). Pathways (1) and (2) showed smaller pathway impact values, whereas pathways (4) and (5) had relatively higher pathway impact values. Thiamine metabolism was significantly different at the six-month stay; however, the pathway impact was small (Figure 4b). No significant difference in the data was observed at the nine-month stay (Figure 4c). Propanoate metabolism was significantly different in the data at the 12-month stay; however, the pathway impact was small (Figure 4d). Detailed data are shown in Figure A2, Figure A3 and Figure A4.

An enrichment analysis was also conducted to evaluate the differences between the depressive and nondepressive episode groups (Figure 5). Five pathways at the three-month stay showed significant differences: (1) D-glutamine and D-glutamate metabolism; (2) nitrogen metabolism; (3) pyrimidine metabolism; (4) glyoxylate and dicarboxylate metabolism; and (4) alanine, aspartate, and glutamate metabolism (Figure 5a). Thiamine metabolism differed significantly at six months (Figure 5b). There was no significant difference at nine months (Figure 5c). Two pathways, (1) propanoate metabolism and (2) pyrimidine metabolism, showed significant differences at 12 months (Figure 5d). The detailed data are shown in Figure A5, Figure A6 and Figure A7.

## 3. Discussion

In this study, the relationship between major depressive episodes and urinary metabolites were investigated during the long-term stays of the 59th JARE in an isolated environment. Of the thirty members, five showed a high PHQ-9 score (≥10) during the 12-month stay. We conducted LC-MS-based metabolomics and compared the urinary metabolites between participants with and without depressive episodes. Changes in metabolites after three-, six-, and nine-month stays were also analyzed to obtain time-series data. These data were analyzed at both the molecular and pathway levels.

A comparison of individual metabolites in the 12-month stay data revealed 11 significantly decreased metabolites. Here, we evaluated the consistency and discrepancy between these metabolites and data reported in other mental illness studies. Among these 11 metabolites, 4-guanidinobutanoate showed the most significant decrease in the depressive episode group (fold change in the depressive episode group/nondepressive episode group, FC = 0.37, *p* = 0.00041). Altered urinary 4-guanidinobutanoate levels have been reported in patients with liver cirrhosis [26]; however, no study has reported its relationship with mental illnesses. This metabolite binds to water and is converted to 4-aminobutanoate (GABA) and urea by the metabolic enzyme guanidinobutyrase. Accordingly, consistent with our 4-guanidinobutanoate data, GABA (FC = 0.64, *p* = 0.065) and urea (FC = 0.82, *p* = 0.12) showed a decreasing trend in the depressive episode group, although it did not decrease significantly.

A significant decrease in 3-methylhistidine levels was observed in the depressive episode group (FC = 0.69, *p* = 0.037). This metabolite is produced by the post-translational methylation of histidine molecules. Serum concentrations of 3-methylhistidine were significantly reduced in depression model rats [27]. Urinary 3-methylhistidine levels in rats with chronic-unpredictable mild stress-induced depression were also significantly decreased [28]. These results are consistent with our data. However, the urinary concentration of this metabolite in patients with postpartum depression was elevated [19], which is in contrast to our data. The urinary concentration of this metabolite is also elevated under fatigue conditions [29]. The concentration of this metabolite may therefore be affected by both mental illness and fatigue conditions.

In the present study, 2-aminoadipate showed a significant decrease in the depressive episode group at both 9 and 12 months. This intermediate metabolite in lysine metabolism has neuroexcitatory actions and regulates kynurenic acid metabolism in the brain [25]. Ornithine levels were also significantly decreased (FC = 0.22, *p* = 0.049). Ornithine is a major intermediate metabolite synthesized from arginine in the arginine–citrulline cycle. Arginine–proline metabolism has been reported to be altered in both patients with MDD and BD [30]. Differences in amino acids have been frequently reported in comparisons between patients with MDD and healthy controls, but no significant differences were found in this study. For example, alanine and glycine levels were elevated in MDD; however, our data showed a decreasing trend without significance (FC = 0.66, *p* = 0.23, and FC = 0.73, *p* = 0.25) [31].

Pathway and enrichment analyses were also performed. In the comparison of the 12-month-stay data, propanoate metabolism was listed as a pathway with significant differences and changes in both analyses (Figure 4d and Figure 5d). This pathway has also been reported as a potential therapeutic target for BD [32]. However, our pathway analysis showed that the pathway impact was low and only β-alanine was mapped to this pathway (Figure A4). Pyrimidine metabolism also showed significant differences (Figure 4d and Figure 5d). Both pathways include β-alanine, an inhibitory neurogenic amino acid. Notably, this metabolite is altered in urinary samples from patients with MDD and BD [33].

A comparison of the 3-month-stay data showed five significantly altered pathways (Figure 4a). These pathways were also ranked high by enrichment analysis (Figure 5a). These pathways included glutamine and glutamate. D-glutamine and D-glutamate metabolism describes the glutamate–glutamine cycle in glial and neuronal cells, with glutamate serving as an excitatory neurotransmitter in the central nervous system [33]. Altered (1) D-glutamine and D-glutamate metabolism and (2) nitrogen metabolism in urine samples have been frequently reported in patients with MDD and BD [31]. In our study, the changes in these pathways were observed only in the 3-month-stay data, and these alterations tended to be smaller in the six-, nine-, and twelve-month data.

No metabolomic study has analyzed the relationship between urinary metabolites and mental health in JARE. However, various teams have investigated the relationship between mental illnesses and metabolic changes in the Antarctic area [34,35]. For example, metabolism related to bone-mass density is disturbed by vitamin D deficiency caused by limited ultraviolet light exposure [36,37]. The season-dependent alteration of constituents of stress-related metabolism, such as cortisol and immunoglobulins, was also observed [38]. The seasonality of plasma insulin and growth hormones has also been reported [39]. The metabolic pathways related to these studies include fat-soluble intermediate metabolites. Alternatively, the current study analyzed only hydrophilic metabolites, which may explain the few overlaps in altered metabolites between our data and the above-mentioned studies [34,35,36,37,38,39].

This study has several limitations. First, the depression screening tool, PHQ-9, was used as an index to divide the participants into depressive and nondepressive episode groups. Second, changes in urinary metabolites over time were examined to investigate changes under long-term isolated stressful conditions. A problem with using metabolites as markers is that many factors may cause unexpected changes in metabolism, such as the diet and lifestyle of individuals. In this study, the cohorts were designed to reduce these effects; participants ate the same diet and stayed in the same environment. However, we did not control for urine sampling timing, presumably because of diurnal variation. Third, the association between urinary catecholamines and depression has been frequently reported [40,41]. Our pretreatment and measurement method could detect dopamine, a catecholamine, in the standard mixture. However, the peak area of dopamine in the urinary samples was below the limits of detection of our method (Figure A8). A similar phenomenon was observed with serotonin. The protocol used herein was able to evaluate a wide range of hydrophilic metabolites within the linearity limits. However, different sample treatments and/or measurement methods should be applied to quantify these catecholamines. Finally, whereas tryptophan, a precursor of these catecholamines [42], was detected at levels above the detection limits, no significant difference was identified between the depressive and nondepressive episode groups.

## 4. Materials and Methods

### 4.1. Subjects and Sample Collections

The 59th JARE stayed at the Showa Station from March 2017 to January 2018. Urinary samples were collected in Japan before the participants visited Antarctica; this served as the baseline data. Participants were required to fill in the following self-administered questionnaires: EPQ-R neuroticism short version and STAI-Y trait anxiety to evaluate personality characteristics, PHQ-9 to evaluate depression episodes, and the Pittsburgh Sleep Quality Index (PSQI) to evaluate sleep quality. In Antarctica, urinary samples and questionnaires were collected every three months (May, Jun., Sep., and Dec. 2018). At each time point, 5 mL of urinary samples were collected in 1.8 mL Nunc Cryotubes (Thermo Fisher, Tokyo, Japan) and were immediately stored at −80 °C. This study was conducted in accordance with the Declaration of Helsinki and was approved by the Institutional Review Board of Tokyo Medical University (protocol code no. SH3712, date 29 May 2017).

### 4.2. Sample Preparation for Metabolomic Analysis

A urine sample (10 μL) was mixed with MeOH containing 149.6 mM NH_4_OH and 1.5 µM internal standards (d_8_-spermine, d_8_-spermidine, d_6_-*N*^1^-acetylspermidine, d_3_-*N*^1^-acetylspermine, d_6_-*N*^1^,*N*^8^-diacetylspermidine, d_6_-*N*^1^,*N*^12^-diacetylspermine, hypoxanthine-^13^C_2_,^15^N, and 1,6-diaminohexsne). Samples were centrifuged at 20,380× *g* for 10 min at 4 °C (MDX-310, Tomy Seiko, Tokyo, Japan), and 90 μL of the supernatant were transferred to a new tube and were vacuum dried for 1 h at room temperature (VC-96W; TAITEC, Saitama, Japan). The samples were reconstituted with 10 μL of 90% methanol (*v*/*v*), and 190 μL of H_2_O were added to the sample solution. The sample was vortexed and centrifuged at 20,380× *g* for 10 min at 4 °C. The supernatant (150 μL) was injected into the LC-MS system. For creatinine determination, a portion of the supernatant was diluted 5000 fold with H_2_O. Firstly, all the samples were measured in four batches. Between every 13 samples, quality control samples were measured and the relative deviation of the peak areas of the detected metabolites were determined. Trimethylaniline *N*-oxide, creatinine, creatine, carnitine, and urea, exhibited peak sizes larger than those of the other metabolites. Thus, secondly, these metabolites in all samples were measured in a single batch with 500-fold dilution. The measurement orders of the samples were randomized, and the standard mixtures were measured consecutively.

### 4.3. Instrument Parameters for Metabolomic Analysis

The instrument parameters of the LC-time-of-flight mass spectrometry (LC-TOFMS) system used in this study were as described previously, with slight modifications [43]. The LC system was an Agilent Technologies 1290 Infinity instrument (Santa Clara, CA, USA), consisting of an autosampler, quaternary pump, and column compartment. MS detection was conducted using an Agilent Technologies G6230B TOF mass spectrometer. In the present study, we used 40 °C as the temperature of the LC columns. The samples were analyzed in positive ion mode. 

### 4.4. Data Processing

Agilent MassHunter Qualitative Analysis software (version B.08.00; Agilent Technologies) was used for data processing. Peaks were extracted according to the retention time and *m/z* values corresponding to those of our standard libraries. Peaks out of the range for a retention time of 0.1 min and *m/z* of 50 ppm were manually integrated for peak area calculations. The peaks below the detection limit were used as 0. The other peaks were within the upper limit of the linearity range. The peak area (no unit) of each metabolite was divided by the area of the corresponding internal standard, resulting in a relative peak area to eliminate the unexpected bias of the sensitivity drift of MS and reduce the effect of ion suppression depending on the relation times. Subsequently, the relative peak area of each metabolite was divided by that of creatinine for the statistical analyses.

### 4.5. Data Analysis

The time course of the quantified PHQ-9 values and urinary metabolite concentrations were obtained. Higher PHQ-9 scores (≥10) and lower scores (<9) were used to classify participants into the major depressive episode and non-major-depressive episode groups, respectively [44]. The difference between these groups was assessed using the Mann–Whitney U test for quantitative values. The *p*-values of each metabolite were adjusted using FDR (Benjamini–Hochberg method) to control for the α error caused by multiple independent tests. Pathway and enrichment analyses were conducted using MetaboAnalyst software (ver 5, https://www.metaboanalyst.ca/, accessed on 1 June 2022) [45]. Quantile normalization was performed for each sample. The metabolite concentration was normalized using log transformation and autoscaling options. The *Homo sapiens* pathways in the Kyoto Encyclopedia of Genes and Genomes (KEGG) (https://www.genome.jp/kegg/kegg1.html, accessed on 1 June 2022) were used for both analyses. A principal component analysis was conducted using all samples.

Statistical analyses and data visualizations were conducted using JMP Pro (ver 16.0.0; SAS Institute, Cary, NC, USA) and GraphPad Prism (ver 9.2.0, GraphPad Software, San Diego, CA, USA). Pathway and enrichment analyses were conducted using MeV TM (ver 4.7.4, Center for Cancer Computational Biology, Dana-Farber Cancer Institute, Boston, MA, USA, http://mev.tm4.org/, accessed on 1 June 2022) [46] and MetaboAnalyst, respectively.

## 5. Conclusions

The psychological changes in 30 participants in the JARE were assessed using PHQ-9 levels. Urinary samples were collected every three months over one year, and metabolomic analyses were conducted. Five participants showed major depressive episodes (PHQ-9 ≥ 10) during the 12-month stay. In the comparison of the data at 12 months, various metabolites showed significant differences; 2-aminoadipate decreased significantly in the depressive episode group at the 9- and 12-month periods, suggesting that such markers may lead to the early detection of depression.

## Figures and Tables

**Figure 1 ijms-24-00943-f001:**
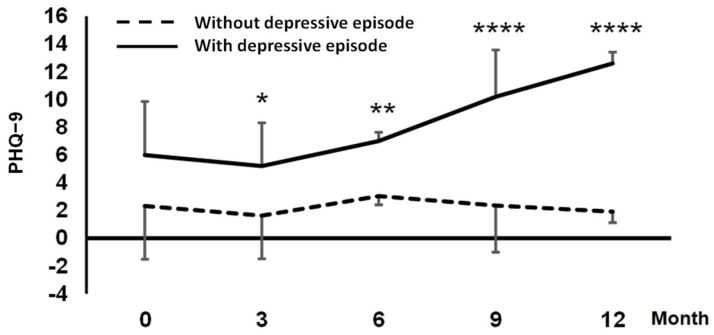
Average and standard deviation of time-series changes in PHQ-9. The solid line indicates the depressive episode group (*n* = 5), and the dotted line indicates the nondepressive episode group (*n* = 25) with PHQ-9 reaching ≥10 at the 12-month stay. Data between the two groups at each time point were compared using Mann–Whitney tests. * *p* < 0.05, ** *p* < 0.01, **** *p* < 0.0001.

**Figure 2 ijms-24-00943-f002:**
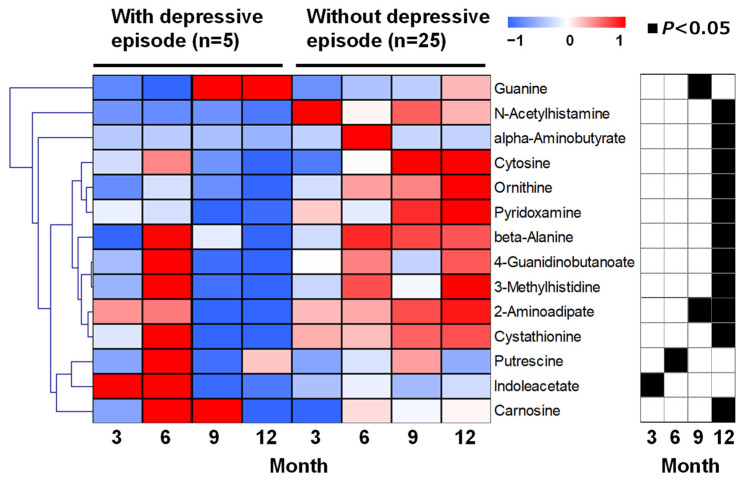
Heatmap of time-series changes in metabolites in the depressive episode and nondepressive episode groups. Mean values were calculated within each group and divided by the mean value for each metabolite to determine the appropriate color. Red and blue indicate values higher and lower than the average, respectively, and white indicates the average. The order of metabolites was clustered using Pearson’s correlation coefficient, and the metabolites with high similarity of change patterns in time series are arranged close to each other. In the graph on the right, black squares indicate data showing a significant difference between the two groups using a Mann–Whitney test (*p* < 0.05).

**Figure 3 ijms-24-00943-f003:**
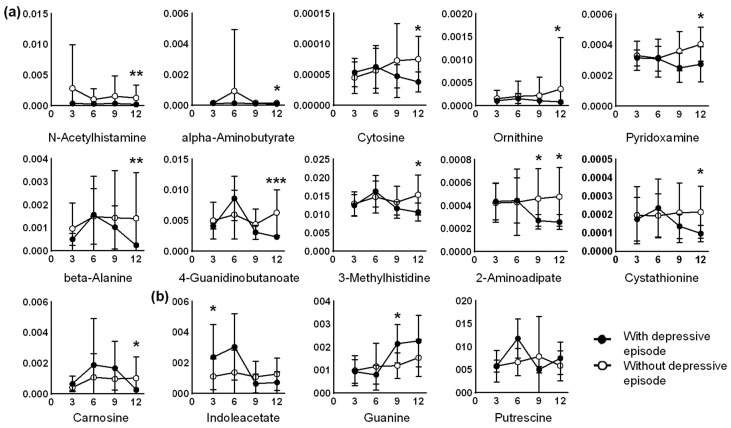
Time-series changes in metabolites that differed between the depressive episode and nondepressive episode groups. Metabolites showing significant differences at the (**a**) 12-month and (**b**) shorter stays are shown. Mean values and standard deviations are shown as plots and error bars, respectively. The X-axis indicates the length of stay (months) and the Y-axis indicates the metabolite concentration divided by the creatinine concentration (unitless). * *p* < 0.05, ** *p* < 0.01, *** *p* < 0.001.

**Figure 4 ijms-24-00943-f004:**
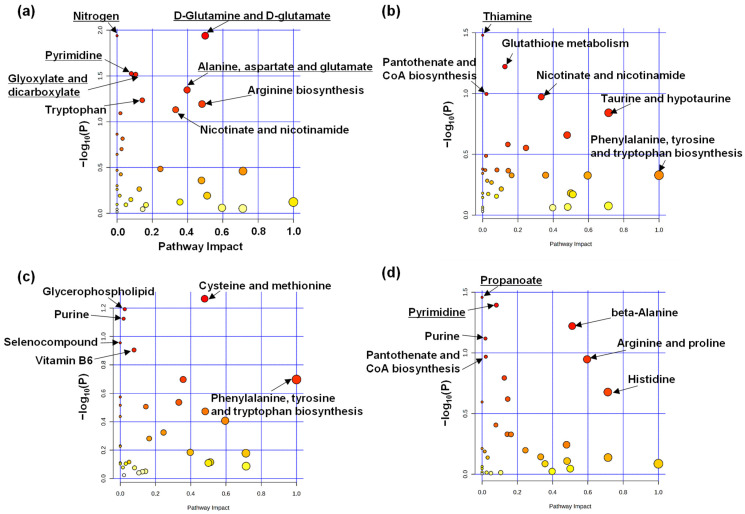
Pathway analysis results between depressive and nondepressive episode groups. Analysis results for (**a**) 3-, (**b**) 6-, (**c**) 9-, and (**d**) 12-month stays are shown. Metabolism is omitted from metabolite names. Underlined pathways indicate significant differences (*p* < 0.05).

**Figure 5 ijms-24-00943-f005:**
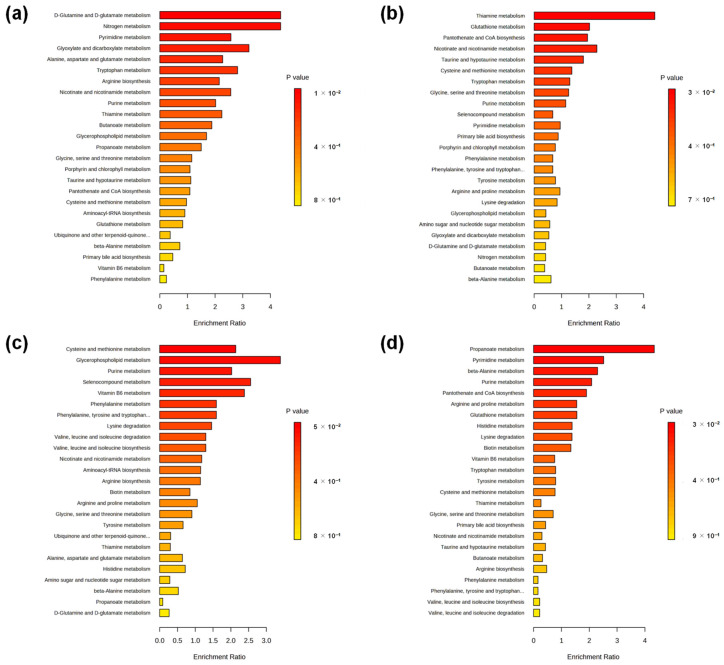
Enrichment analysis between depressive and nondepressive episode groups. In each panel, pathways are aligned according to *p*-values, and the length of each bar represents the enrichment ratio. Analysis results for (**a**) 3-, (**b**) 6-, (**c**) 9-, and (**d**) 12-month stays are shown.

**Table 1 ijms-24-00943-t001:** Subject characteristics.

Characteristic	Average/n/h	Standard Deviation
Age	40.2	9.2
Sex	M 29, F 1	
Education Years	15.3	2.2
Subjective Social Status ^1^	5.5	1.2
Marital Status	Married 18, Unmarried 11	
Presence of Physical Disease	Y 1, N 29	
Presence of Psychiatric Disease	Y 0, N 30	
History of Psychiatric Disease	Y 2, N 28	
Family history of Psychiatric Disease	Y 2, N 28	
Alcohol Drinking	Y 24, N 6	
Smoking	Y 6, N 24	
Overtime work (h/month)		
≤20	19	
≤30	4	
≤40	4	
≤50	2	
Parental Bonding Instrument		
Paternal Care	21.7	6.7
Paternal Overprotection	10.8	5.7
Maternal Care	26.4	5.4
Maternal Overprotection	11.5	6.2
PHQ-9 total scores ^2^	3	3.5
State anxiety of STAI-Y	41.7	9.2
Trait anxiety of STAI-Y	40.1	9.6
Neuroticism (0−12)	3.2	3
Pittsburgh Sleep Quality Index Global Scores	4.5	1.9

^1^ 1 and 10 are the lowest and the highest, respectively. ^2^ Depressive symptoms.

## Data Availability

Not applicable.

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
