# Peer review of "Relationship between Urinary Metabolomic Profiles and Depressive Episode in Antarctica"

_ijms, 2023, doi:10.3390/ijms24020943_

Round 1
Reviewer 1 Report
In this manuscript, Kazuhiko Kasuya, Satoshi Imura, Takashi Ishikawa, Masahiro Sugimoto, and Takeshi Inoue, presented very interesting and important studies, in which the researchers examined changes in the concentrations of many biochemical markers depending on the severity of depressive states, focusing more on the impact of social isolation than on the lack of access to sunlight. There is little research of this type, which is a great asset and novelty in this work. In general, the work is well written and understandable and arouses the interest of the reader. After analyzing the work, only three doubts arise:
1. The secretion of catecholamines (adrenaline, noradrenaline and dopamine) is closely correlated with our emotions. Catecholamines are involved in processes related to concentration, memory, mood, and help cope with stressful situations. Thus, the level of catechol anins also affects the regulation of depressive episodes. The authors of the paper described the method of testing selected biochemical markers in urine. Since urine was the material for the study, why did the authors not include catecholamine metabolites (HVA and VMA) in their studies, which are quite characteristic markers of the level of catecholamines and thus the absence or appearance of depressive symptoms. ??
2. Serotonin and tryptophan levels are also of great importance to determine the severity of depression. Why were these parameters not taken into account?
3. The authors of the paper examined 14 markers by HPLC. From an analytical point of view, this is very interesting. Were all markers tested simultaneously in one run? Maybe it would be worth presenting a chromatogram showing the entire spectrum of analysis of such a number of markers?
Author Response
In this manuscript, Kazuhiko Kasuya, Satoshi Imura, Takashi Ishikawa, Masahiro Sugimoto, and Takeshi Inoue, presented very interesting and important studies, in which the researchers examined changes in the concentrations of many biochemical markers depending on the severity of depressive states, focusing more on the impact of social isolation than on the lack of access to sunlight. There is little research of this type, which is a great asset and novelty in this work. In general, the work is well written and understandable and arouses the interest of the reader. After analyzing the work, only three doubts arise:
- The secretion of catecholamines (adrenaline, noradrenaline and dopamine) is closely correlated with our emotions. Catecholamines are involved in processes related to concentration, memory, mood, and help cope with stressful situations. Thus, the level of catechol anins also affects the regulation of depressive episodes. The authors of the paper described the method of testing selected biochemical markers in urine. Since urine was the material for the study, why did the authors not include catecholamine metabolites (HVA and VMA) in their studies, which are quite characteristic markers of the level of catecholamines and thus the absence or appearance of depressive symptoms. ??
Response: We thank the Reviewer for the positive comments related to our study and for this thoughtful query. We agree with the Reviewer regarding the importance of evaluating catecholamines in relation to depressive symptoms. Indeed, the methods used in our study were capable of detecting catecholamines in the standard mixture. However, the corresponding peaks in urinary samples were smaller than the detection limits of the method. In consideration of the Reviewer’s comments, we recognize this issue as a severe limitation of this study and have accordingly revised the statement of limitations in our manuscript and added Figure A8 (showing the respective peaks) to highlight this concern (lines 284–289 and 417–418).
Lines 284–289:
Our pretreatment and measurement method could detect dopamine, a catecholamine, in the standard mixture. However, the peak area of dopamine in urinary samples was below the limits of detection of our method (Figure A8). A similar phenomenon was observed with serotonin. The protocol used herein was able to evaluate a wide range of hydrophilic metabolites within the linearity limits. However, different sample treatments and/or measurement methods should be applied to quantify these catecholamines.”
Line 417–418:
Figure A8. Representative extracted ion chromatographs of dopamine (m/z = 154.0863, positive mode). A) Standard mixture. B) Urinary sample.
- Serotonin and tryptophan levels are also of great importance to determine the severity of depression. Why were these parameters not taken into account?
Response: We also appreciate this comment. The detection of these metabolites has also been included in the discussion of limitations in the revised manuscript (lines 290–292). Specifically, the serotonin signal was also below the limit of detection for our assay. Tryptophan levels were detectable; however, these did not differ significantly between the depressive and non-depressive episode groups.
Llines 290–292:
Finally, whereas tryptophan, a precursor of these catecholamines [42], was detected at levels above the detection limits, no significant difference was identified between the depressive and non-depressive episode groups.
- The authors of the paper examined 14 markers by HPLC. From an analytical point of view, this is very interesting. Were all markers tested simultaneously in one run? Maybe it would be worth presenting a chromatogram showing the entire spectrum of analysis of such a number of markers?
Response: We thank the Reviewer for the careful consideration of our data presentation. In accordance with the Reviewer’s comment, we added an explanation of the batch measurements and quality control samples used in our study in the revised Material and Methods section (lines 319–324). The deviation of peak area with regard to quality control samples was also added in the revised Results section (lines 130-131). Moreover, examples of chromatographs were added in Figure A8 (lines 417-418).
Lines 319–324:
Firstly, all the samples were measured in four batches. Between every 13 samples, quality control samples were measured and the relative deviation of the peak areas of the detected metabolites were determined. Trimetylalanine N-oxide, creatinine, creatine, carnitine, and urea, which exhibited speak size larger than those of the other metabolites. Thus, secondly, these metabolite in all samples were measured in a single batch with 500-fold dilution.
Lines 130-131:
The deviation of the peak areas among the quality control samples was 9.97 ± 7.09%.
Lines 417-418:
Figure A8. Representative extracted ion chromatographs of dopamine (m/z = 154.0863, positive mode). A) Standard mixture. B) Urinary sample.

Reviewer 2 Report
In this study, the authors have established the Relationship between urinary metabolomic profiles and depressive episode in Antarctica. The study was scientifically sound and conducted with scientific rigour. The time transitions of urinary metabolites established in this study could help in the development of urinary biomarkers to detect subjects with depressive episodes at an early stage.
However, the conceptual framework is missing in this manuscript which is critical to this manuscript and authors are expected to add this into the manuscript to make the introduction of this manuscript theoritically strong and sound.
Author Response
In this study, the authors have established the Relationship between urinary metabolomic profiles and depressive episode in Antarctica. The study was scientifically sound and conducted with scientific rigour. The time transitions of urinary metabolites established in this study could help in the development of urinary biomarkers to detect subjects with depressive episodes at an early stage.
However, the conceptual framework is missing in this manuscript which is critical to this manuscript and authors are expected to add this into the manuscript to make the introduction of this manuscript theoritically strong and sound.
Response: We thank the Reviewer for these positive and valuable comments. According to the Reviewer’s comments, we thoroughly revised the Introduction to clearly present the theoretical background of this study (lines 35–42, 52–56, 64–69, 80-90, 96–102, and 106–107). We also added six new references [1,2,6,8,9,18] to strengthen the theoretical background in the revised manuscript.
lines 35–42:
People are resistant and adaptable to various environmental conditions. In particular, expeditioneers in the Antarctic are exposed to considerable stress through isolation and extreme physical conditions [1], leading to unique psychological effects. In addition to the cold, danger, and hardship, the difficulties associated with individual adjustment to the group constitutes a major stressor [2], with members of Antarctic expeditions undergoing psychological changes resulting from exposure to long periods of isolation and confinement [3]. The psychological effects of polar expeditions in patients under abnormal light, cardiopulmonary, and cold conditions have been frequently investigated [3]
Lines 52–56:
The identification of psychological risk factors that have a negative effect on mood and behavior is desirable for designing efficient preventive approaches to reduce symptoms and generate satisfactory adaptation [6]. Seasonal patterns in depressive symptoms, such as winter-over syndrome, polar T3 Syndrome, and subsyndromal seasonal affective disorder, have been identified [7-9].
Lines 64–69:
Together, these research reports reveal that not only assessing psychological changes but also appropriate early care is necessary to reduce the effects of psychological risks during Antarctic expeditions.
Molecular biomarkers have been identified to predict and quantitatively assess altered mental and immune status.
Lines 80-90:
Nevertheless, the evaluation of novel biomarkers to determine the association with their mental health in the extreme conditions associated with Antarctic expeditions remains necessary.
Recent omics technologies, which enable the simultaneous identification and quantification of thousands of molecules, have been used to explore new biomarkers associated with mental health. Among molecular profiling approaches, the large-scale analysis of metabolites via metabolomics has been recently used for biomarker discovery in the psychiatry field [18]. Owing to the large diversity of the chemical features of metabolites, a single approach cannot profile all metabolites; thus, the combination of individual separation systems prior to detection has been applied.
Lines 96-102:
However, these studies were designed as cross-sectional studies, whereas a longitudinal study is necessary to evaluate the trend of changes in biomarker levels accompanied by mental status in order to realize the early identification of mental changes.
In this study, we aimed to elucidate the urinary metabolite biomarkers that have potential to discriminate subjects with depressive episodes from those without these symptoms.
Lines 106-107:
The individual metabolites and pathway-level changes associated with the transition of depressive episodes were identified.
Added references:
1. Rothblum, E.D. Psychological factors in the antarctic. J. Psychol. 1990, 124, 253-273. doi:10.1080/00223980.1990.10543221
2. Mullin, C.S., Jr. Some psychological aspects of isolated Antarctic living. Am. J. Psychiatry 1960, 117, 323-325. doi:10.1176/ajp.117.4.323
6. Zimmer, M.; Cabral, J.C.C.R.; Borges, F.C.; Côco, K.G.; Hameister, B.d.R. Psychological changes arising from an Antarctic stay: Systematic overview. Estudos de Psicologia (Campinas) 2013, 30, 415-423. doi:10.1590/S0103-166X2013000300011
8. Palinkas, L.A. Effects of physical and social environments on the health and well-being of Antarctic winter-over personnel. Environ Behav 1991, 23, 782-799. doi:10.1177/0013916591236008
9. Reed, H.L.; Reedy, K.R.; Palinkas, L.A.; Van Do, N.; Finney, N.S.; Case, H.S.; LeMar, H.J.; Wright, J.; Thomas, J. Impairment in cognitive and exercise performance during prolonged antarctic residence: effect of thyroxine supplementation in the polar triiodothyronine syndrome. J. Clin. Endocrinol Metab. 2001, 86, 110-116. doi:10.1210/jcem.86.1.7092
18. Shih, P.B. Metabolomics Biomarkers for Precision Psychiatry. Adv Exp Med Biol 2019, 1161, 101-113. doi:10.1007/978-3-030-21735-8_10
